# Improvement of the Antibacterial Activity of Phage Lysin-Derived Peptide P87 through Maximization of Physicochemical Properties and Assessment of Its Therapeutic Potential

**DOI:** 10.3390/antibiotics11101448

**Published:** 2022-10-21

**Authors:** Roberto Vázquez, Antonio Doménech-Sánchez, Susana Ruiz, Julio Sempere, Jose Yuste, Sebastián Albertí, Pedro García

**Affiliations:** 1Centro de Investigaciones Biológicas Margarita Salas (CIB-CSIC), Ramiro de Maeztu 9, 28048 Madrid, Spain; 2Centro de Investigación Biomédica en Red de Enfermedades Respiratorias (CIBERES), 28029 Madrid, Spain; 3Instituto Universitario de Investigación en Ciencias de la Salud, Universidad de las Islas Baleares, 07122 Palma de Mallorca, Spain; 4Instituto de Investigación Sanitaria de les Illes Balears, 07122 Palma de Mallorca, Spain; 5Spanish Pneumococcal Reference Laboratory, Centro Nacional de Microbiología, Instituto de Salud Carlos III, 28220 Madrid, Spain

**Keywords:** phage lysins, antimicrobial peptides, *Pseudomonas aeruginosa*, synergy

## Abstract

Phage lysins are a promising alternative to common antibiotic chemotherapy. However, they have been regarded as less effective against Gram-negative pathogens unless engineered, e.g., by fusing them to antimicrobial peptides (AMPs). AMPs themselves pose an alternative to antibiotics. In this work, AMP P87, previously derived from a phage lysin (Pae87) with a presumed nonenzymatic mode-of-action, was investigated to improve its antibacterial activity. Five modifications were designed to maximize the hydrophobic moment and net charge, producing the modified peptide P88, which was evaluated in terms of bactericidal activity, cytotoxicity, MICs or synergy with antibiotics. P88 had a better bactericidal performance than P87 (an average of 6.0 vs. 1.5 log-killing activity on *Pseudomonas aeruginosa* strains treated with 10 µM). This did not correlate with a dramatic increase in cytotoxicity as assayed on A549 cell cultures. P88 was active against a range of *P. aeruginosa* isolates, with no intrinsic resistance factors identified. Synergy with some antibiotics was observed in vitro, in complex media, and in a respiratory infection mouse model. Therefore, P88 can be a new addition to the therapeutic toolbox of alternative antimicrobials against Gram-negative pathogens as a sole therapeutic, a complement to antibiotics, or a part to engineer proteinaceous antimicrobials.

## 1. Introduction

Phage lysins are currently proposed as an alternative antibacterial therapeutic to be used against multidrug-resistant bacteria [1]. Many clinical trials involving lysins are underway [2,3] and thus the perspectives are good towards a short-term market entry. While the better developed examples of therapeutic lysins are those against Gram-positive pathogens, Gram-negative ones were once deemed rather refractory to the mode-of-action of lysins. Since lysins are enzymes that work by breaking bonds within the cell wall peptidoglycan, the assumption of Gram-negative non-susceptibility was due to the presence of the outer membrane (OM). Such a membrane would act as a natural barrier in Gram-negatives, preventing the lysins from reaching the peptidoglycan from without the cell. This old paradigm has been questioned largely by the introduction of engineered lysins [4].

Collectively, these engineering efforts rely on the same principle: fusing the lysin to a domain able to bind or disrupt the Gram-negative bacterial surface (i.e., the OM or any of its elements). Usually, this additional domain is an antimicrobial peptide (AMP) with the potential to permeabilize the OM. In this context, AMPs have grown increasingly relevant in the general field of proteinaceous antimicrobials development. AMPs are their own class of antimicrobials, with some alleged advantages that make them valuable as novel therapeutics: a slower emergence of resistance, good antibiofilm activity, or even some immunomodulatory properties [5]. Also, being of protein nature, AMPs can be relatively easily designed, tuned, and synthesized. As a drawback, AMPs have been found to be, in some instances, poorly stable in vivo (e.g., due to proteolytic degradation); may present confounding immunological effects; or systemic/local toxicity effects [5].

Some physicochemical properties that enable AMPs to closely interact with biological membranes are reportedly widespread among them, such as a high cationic charge or high hydrophobic moment [6]. This is explained because surface-related interactions are thought to be the fundamental mechanism underlying AMP activity: a positive charge improves interaction with the negatively charged cell surface and a high hydrophobic moment allows for hydrophobic stretches to insert the membranes. Tuning specific properties of AMPs by targeted modifications in their amino acid sequence have allowed for improving AMPs [7,8].

In a similar way to lysins, AMPs reservoir is also broad, comprising many sources among all of life’s domains. The latest literature implies that AMPs can also be derived from phage lysins [9,10,11], particularly from those of phages that infect Gram-negative bacteria, as they typically contain AMP-like subdomains that may have evolved to overcome the many Gram-negative surface barriers, including the OM [12,13,14]. In this regard, our recent results have shown that the prediction of AMP-like elements within lysins is a valid method towards developing membrane-interacting lysins [15]. Using this strategy, the lysin Pae87 was identified and characterized as a prospective antimicrobial agent against *Pseudomonas aeruginosa* and other Gram-negative pathogens. The previously presented results suggested that most of the observed antibacterial effect of Pae87 was due to a nonenzymatic activity enclosed at its C-terminal AMP-like region termed P87 [16]. In fact, P87 itself showed an antibacterial activity of its own that was promising towards developing it as a therapeutic agent of its own.

Therefore, in this work we propose the optimization of P87 antimicrobial activity by means of maximizing features relevant to the interaction with its presumed target (i.e., the biological membranes). This aim was achieved by introducing some modifications in the P87 sequence leading to the increase of its net charge and hydrophobic moment, thus generating the novel synthetic AMP P88. Furthermore, the therapeutic potential of P88 was evaluated by a range of in vitro assays, including: the examination of its activity range against collections of pathogens; the determination of its possible cytotoxic effect; its ability to act synergistically together with antibiotics; and its antibiofilm effect. Altogether, the comprehensive results hereby presented would allow the proposal of P88 as a sole or complement antibacterial chemotherapy, and as a novel part for the engineering of proteinaceous antimicrobials.

## 2. Results and Discussion

### 2.1. Design of an Optimized AMP Based on P87

On the basis that a higher net charge and hydrophobic moment within a proteinaceous structure implies a better chance for it to interact with the Gram-negative surface [12,15], several modifications of peptide P87 (Figure 1a) were proposed to improve its bactericidal activity (Figure 1b). Similar modification-based tuning has been previously reported in the literature [17,18].

The rationale followed was: (i) clustering the positive charges in one side of the predicted alpha-helices of the peptide subtending a relatively “small” angle without dramatically changing the wild-type sequence (hence modifications H15F or I10K); (ii) removing the negatively-charged residue (E25Q); (ii) favoring an asymmetric distribution of polar and nonpolar residues (e.g., T3A, N11A or E25Q). The five incorporated modifications theoretically increased the net charge by ≈ 2 units while also increasing amphipathicity (from 0.63 to 0.81). Given JPred secondary structure prediction, such modifications were not expected to significantly hamper the new peptide P88 helix propensity (Figure 1c). In fact, in the circular dichroism far-UV spectra in the presence of TFE, peaks at ≈ 222 and 208 nm, characteristic of α-helices, were visible at the higher TFE concentrations (Figure 2a). This suggested that P88 retained the ability to form α-helices in the presence of biological membranes.

Assays with the membrane permeabilization probes NPN and SYTOX showed that equimolar concentrations of P88 permeabilized the OM (Figure 2b) and both the outer and the inner membrane (Figure 2c,d) more effectively than P87. These results proved that P88 has a better disruptive interaction with the biologic membranes than the parental P87, and thus the designed modifications accomplished their predicted aim.

### 2.2. Compared Antibacterial Activity of the Novel Peptide P88

To further check whether the enhanced properties of P88 correlated with greater antibacterial ability, the modified peptide was tested against *P. aeruginosa* PAO1 suspensions in comparison with equimolar amounts of P87. Like P87, P88 also exerted a rapid lytic effect when added to a PAO1 suspension (Figure 3a). It is worth noting that P88 lytic kinetics seemed faster than that of P87. While a relevant dependency on the peptide:bacteria ratio was observed for P87 (its maximum activity at 10 µM P87 was achieved with a bacterial dose of 107 CFU/mL or lower), the activity of 10 µM P88 was not hampered with as much cell density as 108 CFU/mL (Figure 3b). This can be interpreted as a 10-fold increase in killing efficiency of P88 in comparison with P87. Likewise, the dose-response curve of P87 up to 20 μM against 108 CFU/mL PAO1 was linear, while P88 rapidly achieved maximum activity at 5 μM (Figure 3c). As for the variation of activity with pH, P88 replicated the trend observed for P87, i.e., its activity was maximized at acidic pHs (Figure 3d). Nonetheless, P88 maintained a remarkable killing activity (around 4 log units) at pHs 6.5–7.0. This observation could be explained by the higher net charge of P88 at the same pH since it has been shown that P87 activity largely relies on a positive charge [16]. The bactericidal spectrum of P88 was the same as P87, pointing to a common mechanism of action, but, again, P88 displayed much higher killing values for all of the susceptible bacteria tested (Figure 3e).

P88 activity was tested against a collection of clinical *P. aeruginosa* strains to check the impact of bacterial genetic variability on its antimicrobial potential. All 30 *P. aeruginosa* clinical isolates were demonstrated to be susceptible to P88 activity, although 10% of them showed a killing effect below 2 log units (Figure 4a). Except for the few less susceptible strains, a relatively low variability was observed in the magnitude of the killing effect (ranging between 4–6 logs). While it would not be unexpected that the origin of isolates determines susceptibility [19], our results showed no significant differences in susceptibility between the strains from different sources (Figure 4b), as also shown for other phage-derived antimicrobials [20].

In addition, a panel of different isogenic mutants of *P. aeruginosa* strain PAO1 exhibiting different degrees of encapsulation or different antimicrobial resistance phenotypes acquired during infection, including overexpression of the chromosomal beta-lactamase AmpC, inactivation of OprD, and absence or overexpression of several efflux pumps was checked. Again, no specific P88 resistance determinants were found (there were no statistically significant differences in P88 susceptibility among the mutant strains; Figure 4c).

Given the clinical and ecological importance of biofilms, P88 was also tested on *P. aeruginosa* biofilms [21]. To that end, a set of *P. aeruginosa* strains were tested for biofilm formation in polystyrene plates.

Figure 5a shows a rather high variability among the crystal violet (CV)-quantified biofilms in the different strains, which is not uncommon [22,23]. Two clinical strains (126.1 and 57.1) were selected to proceed on the basis of their good-biofilm forming capacity and susceptibility to P87 and P88. In both cases, a rather mild disaggregation effect (as evidenced by the CV measurements) correlated with 2–3 log units of killing (Figure 5b,c). P88 exerted the highest killing activity, but below its previously shown activity against planktonic cultures. Given the structure of biofilms, it is to some point expected a hampering to the effectiveness of the different antimicrobials. However, the fact that a relevant bactericidal effect is detected against biofilms of two different clinical *P. aeruginosa* strains is a good prospect towards their application as therapeutics or disinfectants.

In general, the antimicrobial activity of P88 is superior in all parameters tested to that of P87, and thus also from that of the parental enzyme Pae87, which was already somewhat an inferior antimicrobial as compared with P87 [12]. The activity of P88 also falls in the range of other lysin-derived peptides. For example, peptide P307_SQ-8C_, derived from lysin PlyF307, killed around 5 logs of a ~10^6^ CFU/mL *A. baumannii* culture using a concentration of ~10 μM [10]. Some previously reported lysins specifically devised against *P. aeruginosa* seem to perform slightly better than peptide P88, but still in the same concentration/activity range. For example, PlyPa03 or PlyPa91 achieved 5-log killing at around 1.5 μM against strain PAO1, but with a cell density of 10^6^ CFU/mL [24]. P88 is able to achieve a comparable killing effect at 5 μM but with 100-fold more bacterial cells which, knowing that the peptide:bacteria stoichiometry plays a most relevant role in the bactericidal outcome (Figure 3b), we believe is a good killing efficacy outcome.

### 2.3. Cytotoxicity of P88

One of the drawbacks usually adduced for the therapeutic application of AMPs is their possible cytotoxic effect against eukaryotic cells. Indeed, our experiments rendered relatively low IC_50_ values as calculated after examining the viability of epithelial A549 lung cell cultures incubated with different P88 peptide concentrations for 48 h (Figure 6).

While the parental enzyme, Pae87, displayed no remarkable cytotoxic effect at the concentrations tested (Figure 6a), both P87 and P88 were somewhat toxic for the A549 cells (Figure 6b,c). After fitting the dose-response curves (Figure 6 legend), the calculated IC_50_ was 49 μM for P87 and 34.4 µM for P88. Although a cytotoxic effect may hamper the therapeutic potential of the peptides, the positive note is that, while the results presented above showed that P88 was ten times more efficient against bacteria than P87, the fold change in toxicity is just 1.42. The known toxic peptide melittin, included as a positive control in the study, had an IC_50_ of 1.9 µg/mL (Figure 6d). This value (≈0.7 µM) was much lower than those estimated for the Pae87-derived peptides. On the other hand, the observed P88 MICs for PAO1 and the clinical strains 39.5 and 126.1 were, respectively, 20 µM and 50 μM (Table 1). The ratio between MIC and IC_50_ is much more reasonable in the case of P88 than in that of the toxic melittin, which is described to have a MIC against *P. aeruginosa* strains ranging between 64 and 128 mg/L (more than 30 times the IC_50_) [25,26]. Of note, this cytotoxicity might not be an issue if the peptides are purposed for such applications as disinfection or food preservation. It also must be underlined that, although these in vitro results can help determine whether a toxic effect may exist or not, the cytotoxicity assays performed with A549 cells are known to overestimate the actual toxicity that is observed in vivo [27].

### 2.4. Synergy of P88 with Antibiotics

To reduce the effective concentration of P88 and its plausible side cytotoxic effect, experiments analyzing the synergistic activity of the peptide in combination with several antibiotics were conducted. Both the MICs of P88 and several antibiotics were calculated for the standard strain PAO1 and the clinical isolates 39.5 and 126.1 (Table 1). The MIC values for P88 (20–50 µM) seem coherent with previously reported MICs of lysin-derived AMPs, although relatively in the higher end. For example, the MICs of unmodified lysin-derived peptide P307 were above 200 µM, while for the engineered P307_SQ-8C_ they were 14–29 µM [10]. Likewise, the reported MIC values for LysAB2-derived peptides are between 4 and 64 µM [11]. Using the MIC values in Table 1, we performed a checkerboard assay and an end-point viability determination to evaluate different combinations of P88 and the antibiotics, using PAO1 and strains 39.5 and 126.1 as models (Figure 7).

The checkerboard assay demonstrated synergy for ERY, CHL, and TET in the 39.5 strain (Figure 7a), but not for LVX, KAN, or GEN (data not shown). The common criterium to prove synergy of a minimum FICI value of 0.5 or below was met since minimum FICI values were 0.46, 0.44, 0.48, and 0.5 respectively for ERY, CHL, TET, and AZI. Further experiments confirmed synergy in all the strains tested (Figure 7b). Overnight cultures of each strain in the presence of 0.25× MIC of P88 did not display a significant decrease in the viable cell counts. A concentration of 0.25× MIC of each antibiotic caused a 1–3 log units decrease in the case of ERY and CHL and ≈4 log units for TET. However, the combination of a mildly active antibiotic concentration with a peptide concentration without an effect on viability yielded viable counts at least 2 logs below the ones of the cultures treated only with the antibiotic. In fact, in some cases, the viable counts of the peptide-antibiotic combination were below the detection limit of the assay. Therefore, the viability criterium for synergy (an increase in killing effect ≥ 2 logs higher in the combination than in the sum of the effects of each compound alone) was also met. As for the possible synergy mechanism, it is quite straightforward that AMPs (and particularly those which interact with biological membranes) can synergize with antibiotics by enhancing the bacterial cell permeability and, thus, the access of antibiotics to their targets [28]. However, this does not provide a clear explanation of the actual reason why P88 demonstrates synergy with some antibiotics with intracellular targets and not others. In general, the mechanisms of synergy are not well understood, and they may even rely on AMP antibacterial activities other than cell permeabilization, or their particular mode-of-action [29].

The presence of a biological fluid (lysed horse blood), had a slightly detrimental effect on P88 antimicrobial activity, increasing strain 39.5 MIC up to ≈128 μM with 50% lysed horse blood (Figure 7c). The AZI MIC, however, dramatically decreased with the blood concentration (down to 8 mg/L at 50% lysed horse blood), as already reported in the literature [30]. This is explained by an increased cell permeability in complex media, and such a phenomenon is also susceptible to hamper the synergistic cooperation of antibiotic and peptide. Our results in 25% lysed horse blood indeed showed a slight decrease of the synergistic activity, but synergy could still be observed (Figure 7d). A synergistic effect was also observed in the *P. aeruginosa* mouse pneumonia model. While the tested concentrations of AZI (100 mg/mL) or P88 failed to provoke any significant decrease in the bacterial counts recovered from lung tissue as compared with the placebo treatment, we detected a reduction of ≈1 log with the combination at a 0.25× MIC concentration of P88 (Figure 7e). The further reduction of P88 concentration (down to 0.125× MIC) failed to produce a significant bacterial decrease. In conclusion, we may say that synergy provides a possible strategy to therapeutically use P88 with concentrations well below the IC_50_ (the effective concentrations in the presence of subinhibitory amounts of antibiotics are 5.0 μM and 12.5 μM for PAO1 and 39.5/126.1, respectively) and allows for postulating P88 as a molecule able to resensitize bacteria to antibiotics.

## 3. Conclusions

A peptide, P87, from phage lysin Pae87 has been tested as a source for efficient, alternative antimicrobials. To improve its antimicrobial efficacy, a derivative peptide, termed P88, with increased hydrophobic moment and net charge has been designed and proven to possess an enhanced bactericidal and antibiofilm activity. Despite its greatly increased antimicrobial activity, P88 showed a cytotoxic effect similar to that of the parental peptide, and it showed synergistic activity, together with some antibiotics in culture medium, blood-supplemented medium and in a pneumonia mouse model of infection. Therefore, P88 could be further employed as a complement therapeutic agent to commonly used antibiotics or as a part for the further engineering of proteinaceous antimicrobials.

## 4. Materials and Methods

### 4.1. Bacterial Strains and Culture Conditions

Bacterial strains used throughout this work and their culture conditions are those listed elsewhere [15]. P. aeruginosa clinical isolates can be found in [31,32]. *P. aeruginosa* PAO1 isogenic mutants are referenced in [33,34].

### 4.2. Proteins and Peptides Obtention

Pae87 was produced and purified as explained in [15]. Peptides P87 (LNTFVRFIKINPAIHKALKSKNWAEFAKR) and P88 (LNAFVRFKKIAPAIFKALKSKNWAQFAKR) were obtained by chemical synthesis and supplied by GenScript as a freeze-dried powder. They were dissolved in water and concentration was estimated by measuring A_280_, with a molar extinction coefficient of 5500 M^−1^.

### 4.3. Circular Dichroism

Spectra were acquired at 4 °C in 20 mM sodium phosphate buffer (NaPiB), pH 6.5, 150 mM NaCl, using a Jasco J700 spectropolarimeter (Jasco, Tokyo, Japan) equipped with a temperature-controlled holder and at a peptide concentration of 0.1 mg/mL. Far UV spectra were recorded from 260 to 200 nm in a 1 mm path length quartz cuvette. Each spectrum was obtained by averaging 5 accumulations collected at a scan rate of 50 nm/min and 2 s of response time. Buffer spectra were subtracted from protein or peptide spectra and molar ellipticity was calculated. Different concentrations of 2,2,2-trifluoroethanol (TFE) were used to predict the ability of peptides to form secondary structures in the presence of biological membranes.

### 4.4. Cell Permeability Fluorescence Assays

SYTOX assays were conducted with suspensions of PAO1 resting cells (≈10^7^ CFU/mL) prepared from an actively growing culture (OD_600_ ≈ 0.4–0.6) pelleted (3000× *g*, 10 min, 20 °C) and resuspended in 20 mM NaPiB, pH 6.0, 150 mM NaCl, 100 mM sorbitol. 100 µL of this suspension were added to each well of a FluoroNunc 96-well plate together with 5 µM of the probe, and fluorescence was recorded in a Varioskan Flash microplate reader (Thermo Fisher Scientific, Waltham, MA, USA) with an excitation wavelength of 485 nm and 520 nm for the fluorescence detection. Incubation was prolonged until baseline was reached and then 100 µL of the compounds assayed at the proper concentration were added to each well. Then, measurements were prolonged for 60 min at 37 °C. NPN assay was similar, but the bacterial suspension was at 10^8^ CFU/mL culture, and the final probe concentration was 10 µM. The fluorescence recording settings were 350 nm excitation, and 420 nm emission.

### 4.5. Antibacterial Activity Resting Cells Assays

Bioassays were performed by incubating a resting bacterial cell suspension in 20 mM NaPiB (with different modifications as stated at the figure legends), together with the corresponding antibacterial molecule. Resting cell suspensions were prepared by harvesting bacteria at the exponential phase as determined by OD_600_ measurements. The pelleted cells were resuspended in buffer and plated onto a 96-well plate (100 μL per well). Additional 100 μL of the same buffer containing the desired concentration of the antimicrobial was then added and the plate was incubated at 37 °C for 2 h. OD_600_ was optionally monitored during the assay and at the end of the experiment viable cell counts by plating 10-fold serial dilutions were conducted.

### 4.6. Cytotoxicity

A549 cells were cultured in Dulbecco’s Modified Eagle’s Medium, 10% fetal bovine serum. For testing cytotoxicity, 80 μL of A549 cells were seeded at 10^4^ cells/mL in each well of a 96-well plate and grown for 24 h at 37 °C. Then, 20 μL of serial dilutions of the peptides or proteins were added onto the cells. Culture medium was used as a negative control. Incubation was prolonged for 48 h at 37 °C. 3-(4,5-dimethylthiazol-2-yl)-2,5-diphenyl tetrazolium bromide (MTT) was used as an indicator of cell viability adding 20 μL per well from a 2.5 mg/mL stock. Plates were incubated in darkness for 4 h, and MTT reaction was stopped with 100 μL of 10% SDS, 45% dimethylformamide, pH 5.5. Absorbance at 570 nm and 690 nm was recorded using a multiwell plate reader, and the percentage of cell viability was calculated with respect to the 100% signal (OD_570_ − OD_690_) of the negative control.

### 4.7. Minimum Inhibitory Concentrations and Synergy

Minimum Inhibitory Concentrations (MICs) of antibiotics or AMPs were determined according to the Clinical and Laboratory Standards Institute (CLSI) guidelines for *P. aeruginosa* antimicrobial susceptibility testing [35]. Interactions between antibiotics and peptides were tested by the checkerboard assay [36,37]. Such assays were performed in the same microtiter plates using the same medium and incubation conditions as described for MIC determinations. Compounds were usually tested in a range from 1/32 to 2× MIC. The fractional inhibitory concentration (FIC) for each well was calculated (FIC_A_ = C_A_/MIC_A_; where FIC_A_ is the fractional inhibitory concentration of compound A, C_A_ is the concentration of compound A in a given well and MIC_A_ is the MIC of compound A) and the sum of FIC or FIC index (FICI = FIC_A_ + FIC_B_) was used to assess whether the combination was synergistic [38]. Synergy was defined, as usual, by a FICI ≤ 0.5, antagonism by a FICI > 4.0 and no interaction (or mere additive interaction) by a FICI of 0.5–4.0. Checkerboard assays were complemented or substituted by final point viable cell determinations in selected combinations.

### 4.8. Biofilms

*P. aeruginosa* biofilms were prepared by plating 200 µL/well of a 1:100 dilution of overnight cultures in trypticase soy broth (TSB) medium onto a 96-well plate. Cultures were incubated overnight (16–20 h) at 37 °C without shaking. Afterwards, OD_600_ was recorded in a VersaMax multi-well plate reader to evaluate total growth. Supernatants were carefully removed from each well and the biomass attached to the plate surface (biofilms) was washed thrice with 250 µL of sterile distilled water per well. For the disaggregation assay, 250 µL of buffer (20 mM NaPiB, pH 6.0, 150 mM NaCl) were added onto each well containing or the corresponding amount of the antibacterial compound assayed. Incubation was then resumed at 37 °C for 2 h. For quantifying the antibiofilm activity, after removing the supernatants and washing thrice with sterile distilled water, 200 µL of water and 50 µL of 1% crystal violet (CV) were added onto each well. Biofilms were stained for 15 min and then the liquid was carefully removed and biofilms were washed thrice with distilled water. Finally, 200 µL of ethanol were added to solubilize the stained biofilms, which were thoroughly scratched and dissolved into the ethanol by pipetting up and down several times. The amount of biofilm was then estimated by measuring A_595_. Alternatively, some wells were reserved for measuring viable cells, in which case, biofilms were directly dissolved into 200 µL of buffer without prior staining, and 10-fold serial dilutions of each well were plated.

### 4.9. In Vivo Mouse Pneumonia Models

BALB/c female mice (8–12 weeks old) weighing about 20 g that were bred in the institution were used. Infection assays using *P. aeruginosa* 39.5 strain were performed to produce pneumonia. Mice under anesthesia with isofluorane were infected via intranasal inoculation with approximately 1 × 10^7^ CFU/mouse. After 24 h of infection, mice were treated with AZI (100 mg/kg) as previously described [39] and/or sub-MIC concentrations of P88 (12.5 and 6.25 µM, respectively 0.25× or 0.125× MIC) administered intranasally in a total volume of 50 μL of water. Bacterial counts were determined from the lungs homogenized in 1.5 mL phosphate buffered saline after 24 h treatment.

### 4.10. Bioinformatic Analyses

Default methods for data representation implemented in ggplot2 [40] such as Generalized Additive Model (GAM) smoothing were used for summarizing complex experimental data when noted. HeliQuest [41] was used to predict physicochemical properties of interest (hydrophobic moment, net charge, hydrophobicity). Secondary structure of peptides was predicted using JPred [42]. General protein and peptide properties were obtained using the ExPASy ProtParam tool [43].

### 4.11. Statistical Analysis

Statistical analysis was performed either in R or using GraphPad InStat version 5.0 (GraphPad Software). Quantitative differences between experimental conditions were analysed to determine their statistical significance by using *t*-test or ANOVA, followed by the post-tests indicated in the legends to the figures. Unless otherwise stated, presented data are means ± standard deviation of at least three independent replicates.

## Figures and Tables

**Figure 1 antibiotics-11-01448-f001:**
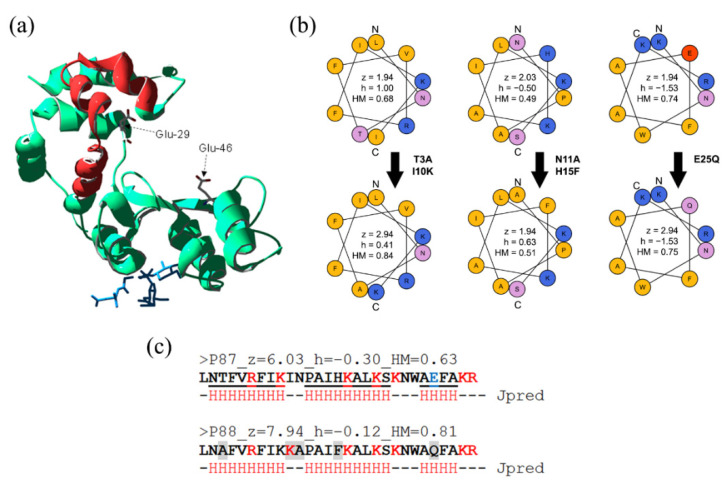
Design of peptide P88. (**a**) Cristal structure of Pae87 [16] (PDB 7Q4T). Peptide P87 is highlighted in red; a bound peptidoglycan fragment (blue) and two catalytic residues (Glu-29 and Glu-46) are presented as spatial references. (**b**) Wheel representation of the three possible α-helices of P87 and the derivative P88. The five P88 modifications with respect to P87 are displayed, as well as the net charge (z), hydrophobicity (h), and hydrophobic moment (HM) calculated for each of the helices. (**c**) P87 and P88 sequences, together with the full sequence net charge, hydrophobicity and HM values, and the JPred secondary structure prediction (a residue marked with an “H” indicates that it is involved in an α-helix). Residues shaded in grey are those mutated in P88.

**Figure 2 antibiotics-11-01448-f002:**
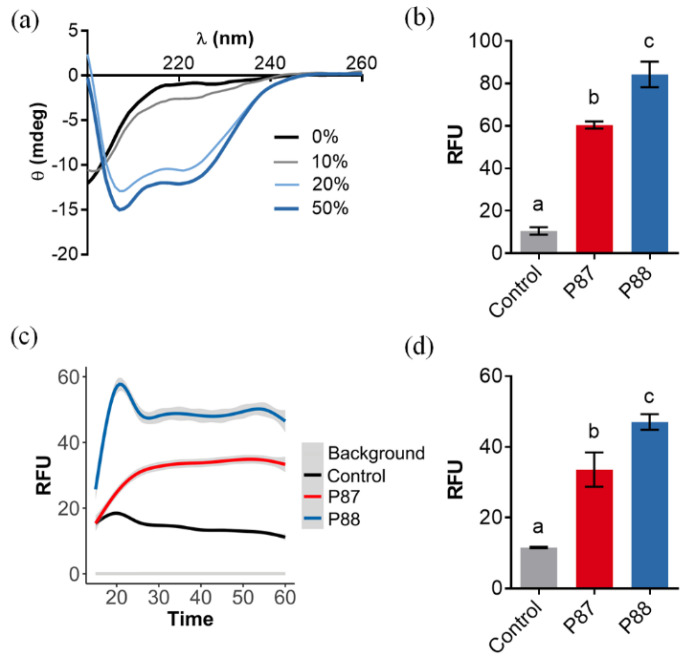
Ability of P88 to form amphipathic helices and permeabilize biological membranes. (**a**) Far-UV circular dichroism spectra of P88 in the presence of different concentrations of TFE. (**b**) Average NPN fluorescence signal after 5 min incubation in the presence of *P. aeruginosa* PAO1. (**c**) Average kinetics of SYTOX fluorescence in the presence of *P. aeruginosa* PAO1. Estimation was based on three independent replicates, mean estimation ± 95% C.I. (grey shade) is shown for each experimental condition. (**d**) Average SYTOX fluorescence at t = 60 min. In (**b**,**d**), means ± sd of three independent replicates are shown and a one-way ANOVA test with Tukey post-test was applied for multiple comparisons. Different letters indicate significant differences.

**Figure 3 antibiotics-11-01448-f003:**
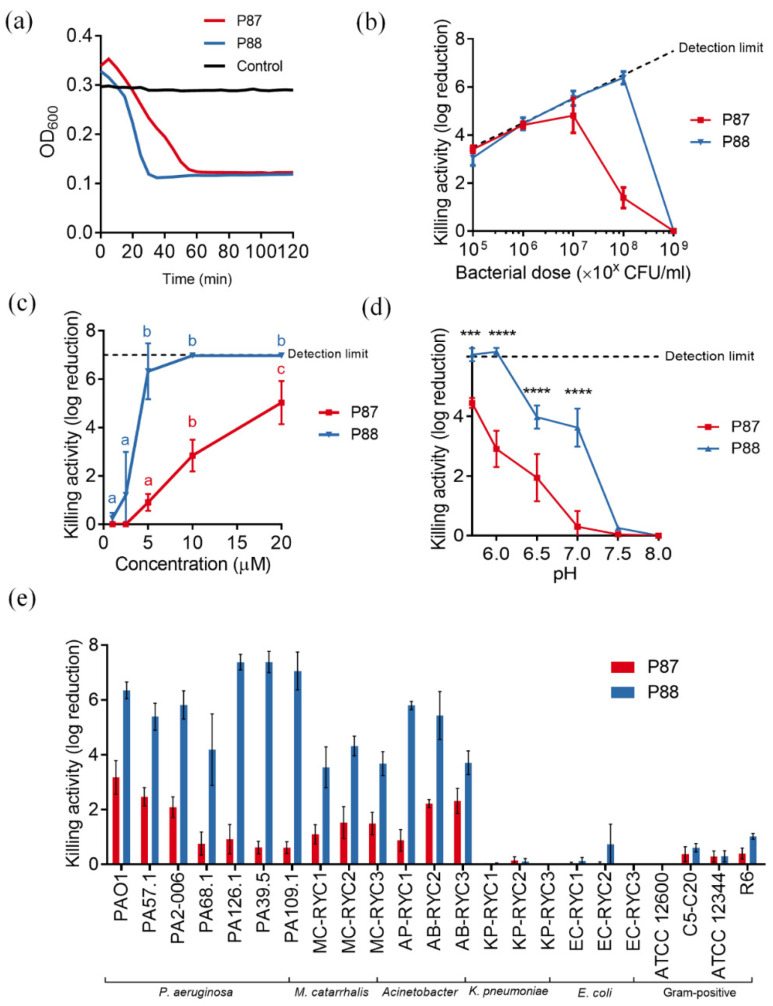
Parameters of P87 and P88 bactericidal activity on different bacteria. (**a**) Bacteriolytic effect. Representative results are shown. (**b**) Peptide:bacteria stoichiometry. (**c**) Dose-response curves. One-way ANOVA plus Tukey post hoc test was applied to compare concentrations within each curve. Different letters indicate significantly different results. (**d**) Variation of killing activity with pH. Two-way ANOVA followed by Dunnett post-test was applied to test significant differences with respect to pH 5.7 (*** *p* ≤ 0.001, **** *p* ≤ 0.0001). (**e**) Bactericidal range. Unless otherwise noted, standard assay conditions were: 20 mM NaPiB, pH 6.0, 150 mM NaCl, ≈10^8^ CFU/mL PAO1 strain, 10 µM of the bactericidal compound, 37 °C, and 2 h incubation.

**Figure 4 antibiotics-11-01448-f004:**
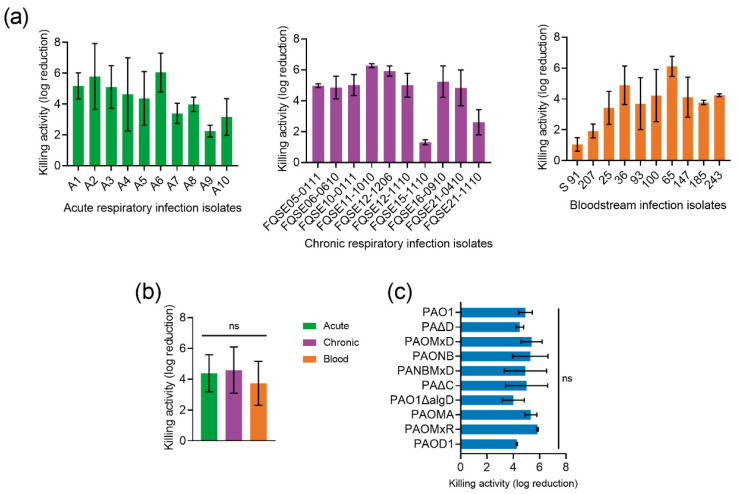
Impact of *P. aeruginosa* genetic variability on P88 activity. (**a**) Bactericidal activity of 10 μM P88 on 10^5^ CFU/mL *P. aeruginosa* clinical strains suspensions (in 20 mM NaPiB, pH 6.0, 150 mM NaCl) incubated for 2 h at 37 °C. (**b**) Mean activity of P88 against the clinical isolates from different origins. One-way ANOVA followed by Tukey post-test was applied to analyze statistical differences. (**c**) Activity of 10 μM P88 on 10^5^ CFU/mL *P. aeruginosa* PAO1 and a collection of isogenic mutants. One-way ANOVA and Dunnett post-test were used to detect statistical differences with respect to the activity against PAO1. ns = Nonsignificant difference.

**Figure 5 antibiotics-11-01448-f005:**
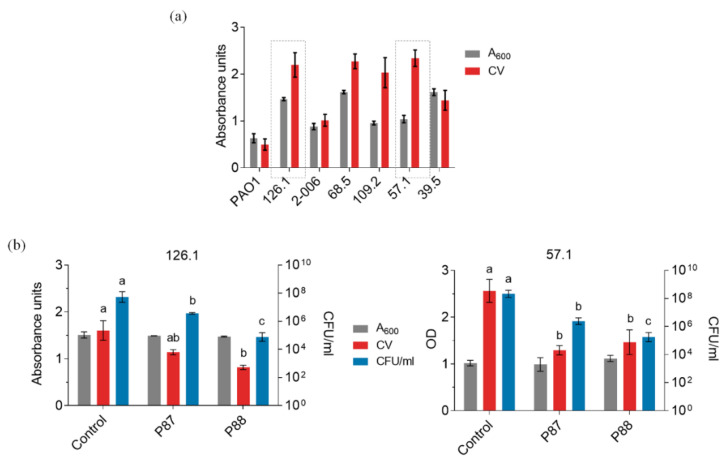
Activity tests of Pae87 derivative peptides against *P. aeruginosa* biofilms. (**a**) Evaluation of the biofilm formation capacity of our available clinical *P. aeruginosa* strains. (**b**) Biofilm disaggregation assays of *P. aeruginosa* strains 126.1 and 57.1 with 10 μM of different compounds applied for 2 h at 37 °C. ANOVA followed by Tukey’s post-test was used to determine significant differences between treatments comparing all-against-all the different conditions (Control, P87, P88) within each type of measurement. Different letters indicate significant differences with a *p*-value of at least ≤ 0.05.

**Figure 6 antibiotics-11-01448-f006:**
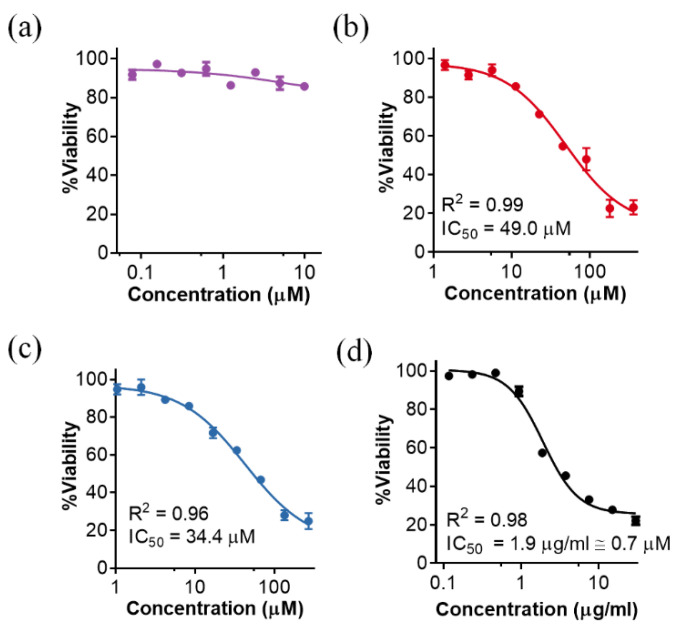
Cytotoxicity of different Pae87-based peptidic compounds on A549 cell cultures. The residual MTT measured viability with respect to an untreated control after 48 h treatment of A549 cells with different concentrations is shown: (**a**) Pae87; (**b**) P87; (**c**) P88; (**d**) melittin as a cell toxicity positive control. Actual data were non-linear fitted to a negative sigmoidal curve (with generalized equation $\mathrm{Viability}=100/(1+10^{\mathrm{HillSlope}\times \left(\mathrm{LogIC}50-\mathrm{Concentration}\right)}$). In this way, IC_50_ was calculated when the fitted sigmoid presented clear top and bottom sections [i.e., (**b**–**d**)]. R^2^ is also presented as a goodness of fit metric.

**Figure 7 antibiotics-11-01448-f007:**
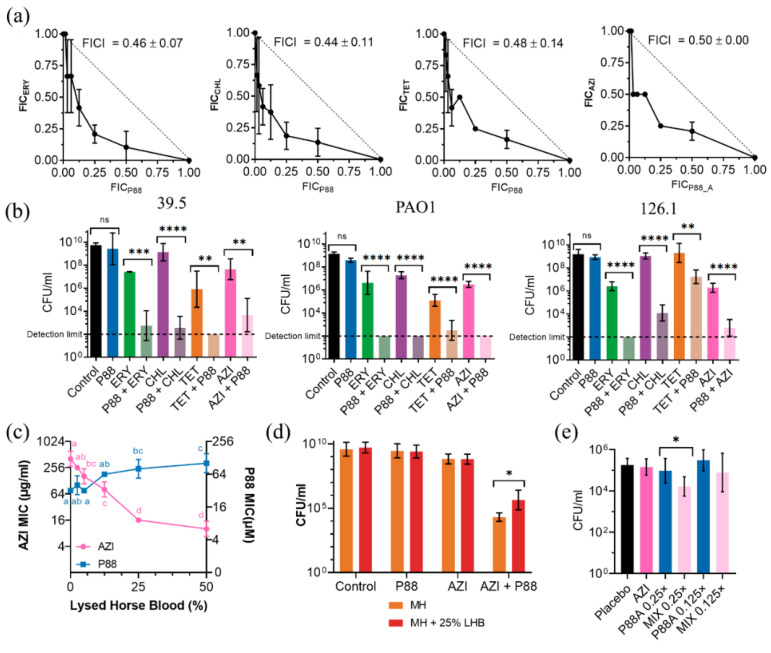
Synergy of peptide P88 with several antibiotics. (**a**) Checkerboard assay isobolograms of different P88-antibiotic combinations against *P. aeruginosa* 39.5 (ERY, CHL, TET, AZI). Mean ± sd of the minimum FICI values are displayed. (**b**) Viable counts of *P. aeruginosa* strains 39.5, PAO1, or 126.1 treated overnight (≈16–20 h) at 37 °C in Mueller-Hinton (MH) medium (initial inocula ≈ 10^5^ CFU/mL) with 0.25× MIC of an antibiotic, P88 or a combination of 0.25× MIC of each, plus the untreated control. One-way ANOVA followed by Tukey post-test was used to determine pairwise significant differences as indicated by the figure (** *p* ≤ 0.01; *** *p* ≤ 0.001; **** *p* ≤ 0.0001; ns = nonsignificant). (**c**) AZI or P88 MIC for *P. aeruginosa* 39.5 determined in MH broth supplemented with different concentrations of lysed horse blood. A two-way ANOVA followed by a Tukey post-test was applied to find statistically significant differences between concentrations within each experimental condition (i.e., comparing the result for the different concentrations of each antimicrobial). Different letters indicate significantly different results. (**d**) Viable counts of *P. aeruginosa* 39.5 treated overnight (≈16–20 h) at 37 °C in MH medium (initial inocula ≈ 10^5^ CFU/mL) with 0.25× MIC of an antibiotic, P88 or a combination of 0.25× MIC of each, plus the untreated control, with or without 25% lysed horse blood. Significant differences according to *t*-test are indicated (* *p* ≤ 0.05). (**e**) Bacterial counts of *P. aeruginosa* 39.5 recovered from the lungs of infected mice and treated with AZI (100 mg/kg), P88 (12.5 µM or 6.25 µM, respectively 0.25× or 0.125× MIC, administered intranasally in 50 μL) or a combination of both. * *p* ≤ 0.05 according to *t*-test. Four to eight independent samples are included per condition.

**Table 1 antibiotics-11-01448-t001:** MICs of P88 and a range of antibiotics ^1^ against *P. aeruginosa* strains.

Antimicrobial	PAO1	39.5	126.1
P88	20 (68) ^2^	50 (170)	50 (170)
LVX	0.25	0.13	0.25
ERY	512	512	256
KAN	16	256	256
CHL	512	512	128
TET	64	1024	32
GEN	0.20	>16.00	2.00
AZI	128	512	256

^1^ LVX = levofloxacin, ERY = erythromycin, KAN = kanamycin, CHL = chloramphenicol, TET = tetracycline, GEN = gentamicin, AZI = azithromycin. ^2^ MICs are given in µg/mL for the antibiotics and in µM for the peptide (with the corresponding value in µg/mL between brackets).

## Data Availability

The raw experimental data are available on reasonable request from the corresponding author.

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
