# Peer review of "Improvement of the Antibacterial Activity of Phage Lysin-Derived Peptide P87 through Maximization of Physicochemical Properties and Assessment of Its Therapeutic Potential"

_antibiotics, 2022, doi:10.3390/antibiotics11101448_

Round 1

Reviewer 1 Report

In this manuscript titled “Improvement of the antibacterial activity of phage lysin-derived peptide P87 through maximization of physicochemical properties and assessment of its therapeutic potential” by Vázquez et al., the authors aimed to evaluate the antibacterial activity of their optimized synthetic antimicrobial peptide (AMP), P88, using several in vitro assays, including biological membrane permeabilization, bacteriolytic effects, and biofilm formation/disaggregation. The title and abstract are appropriate, and the figures are, for the most part, very well presented. However, I have one minor suggestion that the authors should consider addressing in order to strengthen their manuscript prior to publication in Antibiotics, which I have detailed below.

Minor concern:

1.     I was a little confused by the mixed usage of letters and asterisks to indicate significant differences within Figures 3 and 7. I think the authors need to chose one of these methods and be consistent with it throughout the manuscript.

Author Response

  1. I was a little confused by the mixed usage of letters and asterisks to indicate significant differences within Figures 3 and 7. I think the authors need to chose one of these methods and be consistent with it throughout the manuscript.

We appreciate the remark and understand the need for consistency but in this particular case we believe the results are well presented as they are. The choice of using either asterisks or letters is not random and it is explained at the corresponding legends.

  • Letters (Figure 3c, Figure 5 and Figure 7c) are used when statistical comparisons were done within a given experimental condition along the variable displayed in X-axis. In this case, multiple, all-against-all comparisons are performed and therefore many significantly different groups may be established, thus the need for letters describing each of these groups. In this case, the use of asterisks would be, we think, detrimental for the clarity of the figures.
  • Asterisks (rest of the figures) are just used to signal pairwise comparisons, e.g., one peptide against the other or a potentially synergistic peptide+antibiotic mix versus just the antibiotic. Using letters in this case would result rather redundant and would decrease the amount of information conveyed (asterisks also allow to show the actual range of the p-value).

Also, using different description systems may be beneficial towards understanding the type of comparison performed in each figure.

For clarity, some additional explanations have been added to the legends of Figure 5 and Figure 7.

Reviewer 2 Report

The manuscript entitled “Improvement of the Antibacterial Activity of Phage Lysin-derived Peptide P87 through Maximization of Physicochemical Properties and Assessment of its Therapeutic Potential” addresses the construction and the evaluation of P88, an antimicrobial peptide designed for improved antimicrobial activity.

The manuscript is well-written, and the results support the initial hypothesis and conclusions.

I thus recommend the acceptance of the manuscript with revisions.

Major concerns:

Results and discussion

Some of the results are impressive, especially the ones that focus on the antimicrobial activity of P88. However, I do not see any discussion in this section. I miss the comparison of the P88 peptide with endolysins, other peptides, and even bacteriocins. We need some comparison parameters.

The authors do not cite Figure 1a in the text.

Section 2.1: In terms of structure, charge, and functionality, how much does P88 peptide differ and resembles artilysins and other AMPs?

Section 2.2. How P88 peptide antimicrobial activity relates to the antimicrobial activity of other endolysins or AMPs from previously studied?

Sections 2.3 and 2.4 have the same title. I believe you made a mistake in this section heading. One more time, I missed some discussion. We need results from other antimicrobial peptides to compare.

Minor concerns:

Abstract:

Line 21: I wouldn’t use the word “feasible” were far away from replacing or even combining antibiotics with endolysins. Please change it to “promising” or “possible.”

Line 29 to 31: Please put Pseudomonas aeruginosa on italic

Line 32: in vitro is also in italic

Introduction:

The introduction is dense and hard to read. Please divide it into paragraphs; one long paragraph is hard to read and focus on.

In addition, the authors miss the chance to discuss endolysins’ functioning against Gram-negative bacteria broadly. Here’s a review published on Antibiotics that summarizes this section: https://doi.org/10.3390/antibiotics10101143.

The authors missed the chance to talk about the use of endolysins against Gram-negative bacteria lies in the combination of endolysins with outer membrane permeabilizers, such as EDTA. In addition, the authors did not discuss SAR-endolysins. When considering natural endolysins, you forgot to mention SAR-endolysins. Please add a few sentences on those and cite these papers: https://doi.org/10.1007/s12602-022-09948-y / https://doi.org/10.1073/pnas.0400957101 / https://doi.org/10.4014/jmb.1403.03035 / https://doi.org/10.1038/nsmb.1681 / https://doi.org/10.1016/j.resmic.2020.103794

Line 49: Please start a new paragraph on “Collectively”

Line 58: New paragraph on “Since.”

Line 62: “in silico” also in italic.

Author Response

Some of the results are impressive, especially the ones that focus on the antimicrobial activity of P88. However, I do not see any discussion in this section. I miss the comparison of the P88 peptide with endolysins, other peptides, and even bacteriocins. We need some comparison parameters.

The authors do not cite Figure 1a in the text.

Figure 1a is cited at the first paragraph in the section 2.1.

An additional, comparative discussion has been added at the end of the section 2.2 using our prior publication on the parental enzyme Pae87 and previous reports of lysins and lysin-derived AMPs.

Section 2.1: In terms of structure, charge, and functionality, how much does P88 peptide differ and resembles artilysins and other AMPs?

Regarding other AMPs, P88 has an above-the-average net charge per residue (NCPR = 0.27, versus the median NCPR = 0.1 calculated on a set of AMPs from the Antimicrobial Peptide Database as described here: https://doi.org/10.3389%2Ffmicb.2021.660403) and an average hydrophobicity (-0.12 vs a median slightly above 0.0, same source as above). The exact function of AMPs, from a mechanistic point of view, is a complex matter as several different mechanisms are proposed for them. But, in general terms, both P87 and P88 are considered to be rather “canonical” AMPs in the sense that they both show alpha-helix propensity only in the presence of a biological membrane proxy (see Figure 2 in this manuscript and https://doi.org/10.1107%2FS2059798322000936) and they show proven membrane permeabilization activity.

When comparing with lysins, the right framework to do so, we think, would be that of the consideration of Gram-negative outer membrane as a barrier to lysin activity. From this perspective, both peptides perform better than most wild-type lysins are considered to against Gram-negatives, as they show a clear lytic effect which is only exerted by most wild-type lysins in the present of membrane-permeabilizing agents. The physicochemical features of P88 are definitely above the average values for lysins, as shown for example here: https://doi.org/10.1128/JVI.00321-21.

As for artilysins, them being a composite of lysins and AMPs, the key point of comparison with P88 would perhaps be the kind of AMPs often found in functional artilysins (besides a mere activity comparison, which can already be found in the additional sentences added at the end of the section 2.2). Many artilysins contain peptides similar to the AMP SMAP-29, which are, as P88, alpha-helical peptides. The main difference between both is their length: P88 is 2-3 times larger than the normally used AMPs in reported artilysins. Therefore, a key point for further P88 improvement may be minimizing its sequence. The activity of artilysins is potentially more efficient than that of AMPs due to the synergistic effect of a nonenzymatic and an enzymatic element, although the range of activities reported for artilysins is comparable to that of P88, as described in the aforementioned section 2.2.

Section 2.2. How P88 peptide antimicrobial activity relates to the antimicrobial activity of other endolysins or AMPs from previously studied?

See new paragraph at the end of section 2.2.

Sections 2.3 and 2.4 have the same title. I believe you made a mistake in this section heading. One more time, I missed some discussion. We need results from other antimicrobial peptides to compare.

Title of section 2.4 changed.

A more focused comparison with other lysin-derived peptides has been added.

Minor concerns:

Abstract:

Line 21: I wouldn’t use the word “feasible” were far away from replacing or even combining antibiotics with endolysins. Please change it to “promising” or “possible.”

Ok.

Line 29 to 31: Please put Pseudomonas aeruginosa on italic

Ok.

Line 32: in vitro is also in italic

 Ok.

Introduction:

The introduction is dense and hard to read. Please divide it into paragraphs; one long paragraph is hard to read and focus on.

The introduction has been slightly modified and divided in paragraphs to make reading easier.

In addition, the authors miss the chance to discuss endolysins’ functioning against Gram-negative bacteria broadly. Here’s a review published on Antibiotics that summarizes this section: https://doi.org/10.3390/antibiotics10101143.

We thank the reviewer for the suggestion. We ourselves have dealt with the topic extensively (e.g. https://www.frontiersin.org/articles/10.3389/fimmu.2018.02252/full, https://doi.org/10.3389/fmicb.2021.660403,  https://doi.org/10.1128/JVI.00321-21). We are thus aware of the many ramifications (small-molecule membrane permeabilizers co-administration, using high pressure treatments, intrinsically active lysins, the many engineered variants…) and we believe that a fair -but brief- discussion of these topics is already present in the Introduction with the -shallow- depth required by the specific scope of the manuscript, which is not discussing the relationships between Gram-negative-targeting lysins and their targeted bacteria, but the engineering of a lysin-derived peptide. Prior work to which this manuscript refers already covers these other related topics.

Reviewer 3 Report

The article "Improvement of the Antibacterial Activity of phage lysin derivied peptide P87 through maximization of physicochemical properties and assessment of its therapeutic potential" explains the optimization of antibacterial activity associated with peptide P87 through different physicochemical properties for enhancing its therapeutic potential.

Major comments:

1. The limitations of peptides for invitro as well as invivo applications must be explained. 

2. The peptides are supposed to be short lived in invivo environment and must be explored for its invivo potential and/or efficiency.

3. The comparative differences associated with lysins and their derived peptides must be explained to demonstrate the efficiency of the studied peptide(s) under consideration. Structural properties can be explained/compared to demonstrate its effectiveness as compared with other reported studies.

4. Conclusion must explain the reason behind the improvement and/or optimization of phage lysin derived optimization of peptide 87

Minor comments:

1. Minor English language and spell check is required.

2. Abbrivations must be rechecked and/reported accordingly

Author Response

  1. The limitations of peptides for invitro as well as invivo applications must be explained. 

This is now briefly discussed at the introduction.

  1. The peptides are supposed to be short lived in invivo environment and must be explored for its invivo potential and/or efficiency.

We agree with the reviewer: one of the potential drawbacks of AMPs is their (alleged) short in vivo half-life. Further developments in this direction should be one of the future targets, but, thus far, we think the aims proposed for the current manuscript have been met.

  1. The comparative differences associated with lysins and their derived peptides must be explained to demonstrate the efficiency of the studied peptide(s) under consideration. Structural properties can be explained/compared to demonstrate its effectiveness as compared with other reported studies.

As also suggested by reviewer #2, this has been included as additional sentences in paragraphs of the sections 2.2 and 2.4.

  1. Conclusion must explain the reason behind the improvement and/or optimization of phage lysin derived optimization of peptide 87

 This has been included at the conclusions.

Minor comments:

  1. Minor English language and spell check is required.
  2. Abbrivations must be rechecked and/reported accordingly

The “Abbreviations” section has been removed as each abbreviation used throughout the manuscript is already defined the first time it appears at the main text.